# Gravitational scaling analysis on spatial diffusion of COVID-19 in Hubei Province, China

Yanguang Chen *, Yajing Li, Shuo Feng, Xiaoming Man, Yuqing Long

Department of Geography, College of Urban and Environmental Sciences, Peking University, Beijing, P.R. China

* chenyg@pku.edu.cn

**Data Availability Statement:** All relevant data are within the paper and its Supporting Information files.

## Abstract

The spatial diffusion of epidemic disease follows distance decay law in geography and social physics, but the mathematical models of distance decay depend on concrete spatio-temporal conditions. This paper is devoted to modeling spatial diffusion patterns of COVID-19 stemming from Wuhan city to Hubei province, China. The modeling approach is to integrate analytical method and experimental method. The local gravity model is derived from allometric scaling and global gravity model, and then the parameters of the local gravity model are estimated by observational data and least squares calculation. The main results are as below. The local gravity model based on power law decay can effectively describe the diffusion patterns and process of COVID-19 in Hubei Province, and the goodness of fit of the gravity model based on negative exponential decay to the observational data is not satisfactory. Further, the goodness of fit of the model to data entirely became better and better over time, the size elasticity coefficient increases first and then decreases, and the distance attenuation exponent decreases first and then increases. Moreover, the significance of spatial autoregressive coefficient in the model is low, and the confidence level is less than 80%. The conclusions can be reached as follows. (1) The spatial diffusion of COVID-19 of Hubei bears long range effect, and the size of a city and the distance of the city to Wuhan affect the total number of confirmed cases. (2) Wuhan direct transmission is the main process in the spatial diffusion of COVID-19 in Hubei at the early stage, and the horizontal transmission between regions is not significant. (3) The effect of spatial lockdown and isolation measures taken by Chinese government against the transmission of COVID-19 is obvious. This study suggests that the role of urban gravity (size and distance) should be taken into account to prevent and control epidemic disease.

## 1 Introduction

Geospatial diffusion is governed by certain scientific laws, which can be described by mathematical language. The theory and models of spatial diffusion have been initially developed in the period of quantitative revolution. Following distance decay law, the diffusion process is

**Funding:** This research was sponsored by the National Natural Science Foundation of China (Grant No. 41671167).

**Competing interests:** The authors have declared that no competing interests exist.

related to spatial interaction [1–4]. Generally speaking, spatial diffusion is directly proportional to size and inversely proportional to distance, which reflects how gravity take effects in geographical systems. The gravity models include two types of decay processes: one is spatial decay based on distance, the other is hierarchical decay based on size [5]. Distance decay tells us that the place far from the source is less influenced than the place nearby, while hierarchical decay tells us the place which has a larger size tends to be more influenced than the place whose size is significantly smaller than the source. Spatial diffusion and hierarchical diffusion are two main characteristics of geographical diffusion processes [4, 6]. The spread of animals and plants, the dissemination of diseases, and the diffusion of new technologies are all subject to certain spatial interaction laws [4, 7]. The diffusion of epidemics reflects a type of spatial processes in geographical evolution. Once an epidemic occurs in a big city, it will spread from the center to the surrounding area and from big cities to small towns. For geographers, it seems that all above is basic knowledge which is unnecessary to be discussed. However, the concrete decay pattern and parameters of the spatial-temporal change of a specific epidemic cannot be explained in general terms. Only through calculation and analysis based on observation data, can we disentangle truth from falsehood step by step. More studies should be made on the rules behind the epidemic for putting forward effective countermeasures for the prevention and control of epidemic diseases.

A complete dataset can help to develop new models or verify existing mathematical models. Novel coronavirus pneumonia broke out in Wuhan in January 2020, and the COVID-19 rapidly spread from Wuhan to the rest region of China with the help of Spring Festival travel rush. The space-time characteristics and mechanism behind the diffusion process are worth exploring. The new data of confirmed cases can be used to study the mathematical laws of geographical spatial diffusion. Taking Hubei Province as a study area and Wuhan as the center of spatial diffusion, we research the spatial and temporal characteristics of COVID-19 spread in Hubei Province by means of gravity models and spatial autoregressive model in this paper. The effects of city size and spatial distance on the spread of COVID-19 are investigated. Where research methods are concerned, both analytical method and experimental method are employed to make models. A series of concepts including allometric scaling, fractal dimension, and spatial autoregression are introduced into classical models. Gravity models are used to analyze the characteristics of core-periphery vertical diffusion from Wuhan, and spatial autoregression is used to investigate whether there is effect of the horizontal cross influence between different cities except for Wuhan. The goal of this paper is to reveal the geographic spatial regularity of the spread of COVID-19. In Section 2, the local gravity model is derived from the global gravity model with the help of the allometric scaling, and then the spatial autoregression term is introduced into the logarithmic linear form of the gravity model. In Section 3, the parameters of the local gravity model and the mixed gravity model comprising autoregressive term are estimated by using least squares calculations. The main calculation results and the corresponding statistical analyses are shown in this part. In Section 4, the chief related problems of the research are discussed. Finally, the discussion is concluded by summarizing the main viewpoints of this study.

## 2 Models and datasets

### 2.1 Global geographical gravity models

The geospatial diffusion obeys the laws of distance decay. In different situations, distance decay patterns can be described by different functions [3, 8–10]. The number of diffused objects usually follow negative power-law whose power exponent varies between 1 and 2 [4]. However, when considering infectious diseases, it is not enough to consider the inverse

relationship between the number and distance only, but also the hierarchical effect caused by population sizes of human settlements. Thus, the gravity models should be taken into consideration. There are two types of gravity models. One is the global gravity models, which describes the attraction between any two places within a region. The flow and distance can be expressed by matrices in the model. The other is the local gravity model, which describes the attraction between one single place and other places around. The flow and distance are represented by vectors. In this paper, we mainly make use of the local gravity model, but we must start from the global gravity model so as to explain it more clearly. A pair of dual mathematical expressions are required to describe the global gravity relation [5]. This pair of mathematical formulae can be expressed as

$$T_{ij} = KP_i^u P_j^v r_{ij}^{-\sigma}, \tag{1}$$

$$T_{ji} = KP_i^v P_j^u r_{ij}^{-\sigma}, \tag{2}$$

in which, $T_{ij}$ is the origin flow, i.e., the outflow from the source, $T_{ji}$ is the received flow, i.e., the inflow to the destination, $P_i$ is the size of place $i$, $P_j$ is the size of place $j$, $r_{ij}$ is the distance from $i$ to $j$, and $u$, $v$, $\sigma$ are the calibration parameters [11–13], which are essentially cross scaling exponents. The above models are essentially a fractal gravity model whose parameters can be explained by fractal theory [5, 14]. By taking double logarithms of both sides of Eqs (1) and (2), the multiple linear regression analysis can be carried out based on the least square method.

## 2.2 Allometric scaling and local gravity model

The local gravity model can be derived and used to describe the core-periphery relationship in geographical analysis. Mathematical models fall into two categories: mechanism models and parametric models [15]. Correspondingly, the approaches of mathematical modeling can be divided into two types: analytical method and experimental method [16]. In this study, the analytical method is utilized to derive the local gravity model, and the experimental method is employed to verify the models by means of observational data (see Section 3). The allometric scaling relation between passenger flow and case transmission quantity can be treated as basic postulate for the mathematical derivation. In fact, if only one of Eqs (1) and (2) is adopted, the local gravity model is actually used. Mackay once used the local gravity model to study the telephone connection between Montreal and its surrounding towns in Canada [17]. The local gravity model for passenger flow can be expressed as Eq (1), which can be simplified in form: one of the two ranks $i$ and $j$ can be removed. Now, we need a local gravity model for case transmission quantity of COVID-19. Generally speaking, the relationship between the passenger flow of cities and the case transmission quantity is a power law [5, 18]

$$N_{ij} = \mu T_{ij}^{\alpha}, \tag{3}$$

where $N_{ij}$ is the case transmission quantity associated with passenger flow $T_{ij}$, $\mu$ is the proportionality coefficient, and $\alpha$ is the scaling exponent ($i$, $j$ = 1,2,3,...,$n$, where $n$ is the number of places). This is essentially an allometric relation, which is proved to be a priori relation [19]. Eq (3) can be converted into a double logarithmic linear relationship. In geographical studies, allometric relationships are common [20]. The effectiveness of Eq (3) can be verified by the analysis of observation data. Substituting Eq (3) into Eq (1) yields

$$N_{ij} = \mu (KP_i^u P_j^v r_{ij}^{-\sigma})_{ij}^{\alpha} = \mu K^{\alpha} P_i^{\alpha u} P_j^{\alpha v} r_{ij}^{-\alpha \sigma}. \tag{4}$$

Based on the core-periphery relationship, the place $i = 0$ can be taken as the central city, and then $P_i$ can be regarded as a constant $P_0$. So Eq (4) can be simplified as

$$N_j = \mu K^\alpha P_0^{\alpha u} P_j^{\alpha v} r_j^{-\alpha \sigma} = \eta P_j^v r_j^{-\beta}. \tag{5}$$

This is a typical local gravity model. The parameters of Eq (5) can be expressed as below:

$$\eta = \mu K^\alpha P_0^{\alpha u}, v = \alpha v, \beta = \alpha \sigma, \tag{6}$$

where $\eta > 0$ is the rescaled gravity coefficient, $\beta > 0$ is the rescaled distance exponent, and $v > 0$ is the rescaled size calibration parameter. Since both $\beta$ and $v$ are greater than 0, the positive sign (+) means a positive influence of city sizes, and the negative sign (-) means a negative influence of distance. Both distance exponent and size exponent are related to fractal dimension [5].

## 2.3 Integrated model of gravity and spatial auto-regression

Spatial diffusion is supposed to be a network process, including vertical spread and horizontal spread. The local gravity model discussed in subsection 2.2 mainly describes the vertical transmission, without considering the transverse diffusion. If only the diffusion of COVID-19 from Wuhan to other cities in Hubei Province is taken into account, the number of confirmed cases is directly proportional to the population size of the city, and inversely proportional to a certain power of the distance to Wuhan. Whether there is horizontal diffusion between other cities except Wuhan is ignored. In a word, the local gravity model only describes vertical diffusion effect—the relationship between Wuhan and other cities without considering horizontal diffusion effect—the connection between other cities except Wuhan. Such a method is inevitably being questioned: why the cross correlation of other cities can be ignored? In order to consider the horizontal relationship, the idea of spatial auto-regression can be introduced. Spatial auto-regression method is based on spatial auto-correlation idea. Taking natural logarithms on both sides of Eq (3) yields a two-variable linear form

$$\ln N_j = \ln \eta + v \ln P_j - \beta \ln r_j. \tag{7}$$

Assuming that COVID-19 spread between other cities except Wuhan, we should consider the spatial self-influence which can be described by the process of spatial auto-regression. By adding autoregressive term to Eq (7), we have

$$\ln N = \ln \eta + v \ln P - \beta \ln r + \rho W \ln N, \tag{8}$$

where $\ln N$ corresponds to the vector of logarithm of human number infected by COVID-19, $P$ corresponds to the vector of urban population size, $r$ is the distance vector, $W$ is the weight matrix based on spatial distance, $\ln \eta$ is the logarithm of gravity coefficient, $v, \beta, \rho$ are regression coefficients, and among them $\rho$ is the spatial autoregressive coefficient. Although the expression of vectors in Eq (8) is not mathematically standard, it is simple, intuitive and easy to understand. The significance level of spatial auto-regression reflects the effectiveness of the powerful spatial isolation measures taken by the government of China. If the autoregressive coefficient $\rho$ is significant, it indicates that there is a horizontal cross relationship between cities except Wuhan at a certain degree of confidence, otherwise the cross relationship is weak and the impact can be ignored. The weak cross transmission of COVID-19 between the cities outside Wuhan can prove the effect of spatial isolation measures.

## 2.4 Data sources and algorithms

The aim of this paper is at revealing the geographical mathematical regularity of spatial diffusion COVID-19. The study area includes the whole area of Hubei Province, China. The

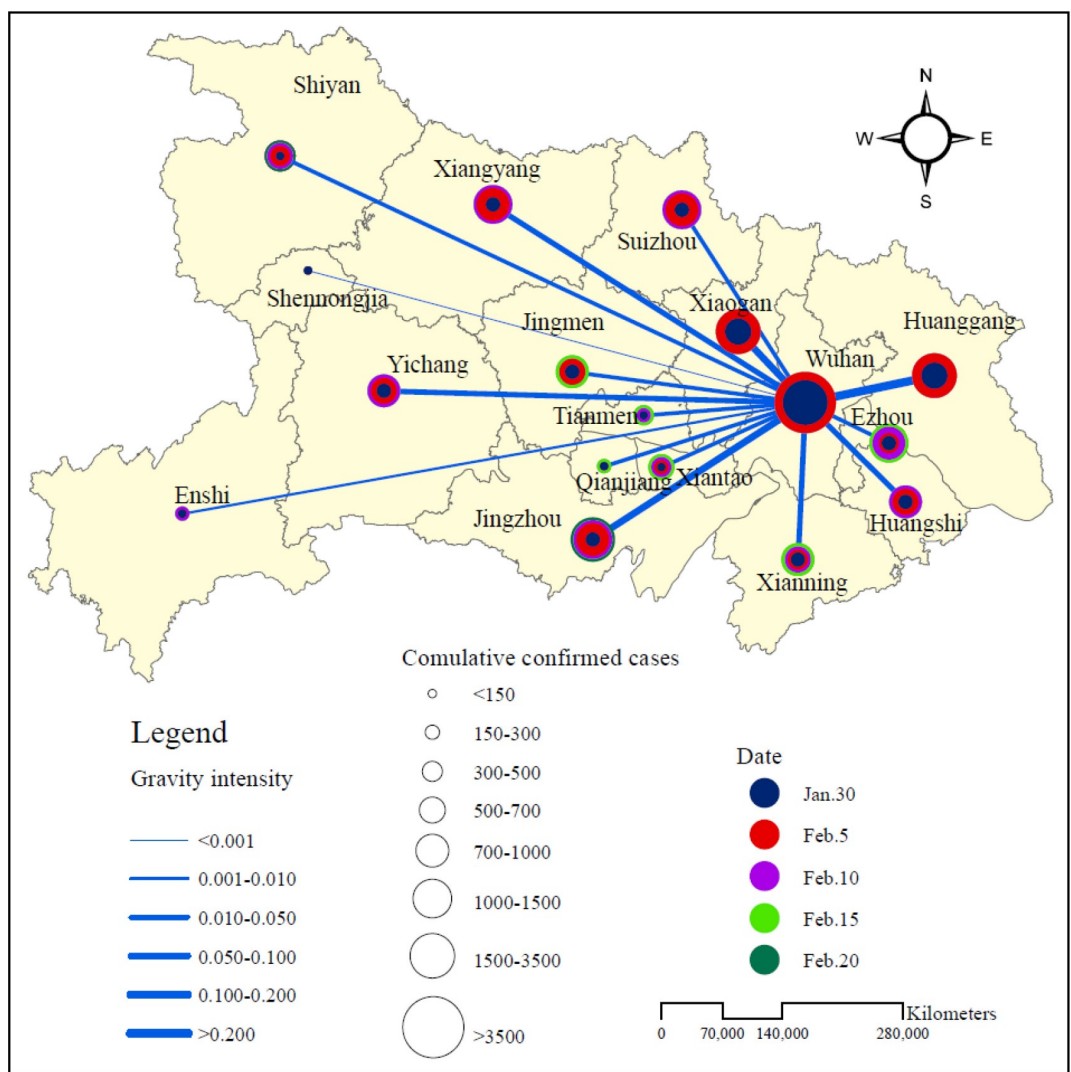

**Fig 1. Local gravity intensity and epidemic diffusion between Wuhan and other regions in Hubei Province (2020).** As shown in the figure, five representative dates are selected to illustrate the scale of the epidemic diffusion. The size of circles represents the scale. The gravity intensity is expressed according to the results calculated from the data of February 20, and the flow intensity is expressed through the line width. The line segments illustrate the vertical diffusion from Wuhan to other cities in Hubei. The horizontal diffusion relationship between other cities outside Wuhan are not shown. The vertical diffusion is described by local gravity model, and the horizontal diffusion is tested by spatial autoregressive analysis.

geographical elements involve the capital of Hubei Province, Wuhan, and all prefecture level cities as well as the interurban road network (Fig 1). According to the local gravity model and auto-regressive model, three sets of data are in need in this study. The first is the number of COVID-19 confirmed cases, which represents the associated variables; the second is the urban population, which represents the size measurement; the third is the traffic mileage between cities, which represents the distance variable and spatial measurement. The confirmed cases number of COVID-19 in prefecture-level cities in Hubei came from the daily report announced by National Health Commission of the People's Republic of China, and the population size of cities was extracted from China City Statistical Yearbook. As for the interurban distances by road, two datasets are chosen. One is the highway mileage table included in Hubei Highway Atlas, which basically provides us with the distance between each prefecture-level

**Table 1. Main data sources used in this study and related explanation.**

| Measurement | Source | Date | Advantage | Disadvantage |
|---|---|---|---|---|
| **Confirmed cases of COVID-19 in Hubei Province** | National Health Commission of the People's Republic of China | January 28 to February 23, 2020 | Cross-sectional data at multiple time points | The cut-off time of some cities is different from the majority |
| **Distance by road (I)** | Hubei highway Atlas | Published in 2012 | Diffusion process can be well explained | Lack of complete matrix |
| **Distance by road (II)** | Measured with GIS technology by authors | February 2020 | Complete matrix of distance | Only reflects the distance covered by private cars which may be different from bus route |
| **Urban population size** | China City Statistical Yearbook 2018 | End of 2017 | Generally reflects the scale of urban system in each region of Hubei province | The data collected two years ago are not real-time statistics |

**Note:** Interurban road distance 1 is obtained from the website "onegreen" (http://www.onegreen.net/maps/). The interurban road distance 2 is obtained with the help of distance analysis tool of DataMap for Excel (AMAP version)—based on "the shortest distance" path planning strategy and the "self-driving" commuting strategy. The latter data were obtained on February 14, 2020.

city in Hubei Province and Wuhan. The dataset of the distances between prefecture level cities is incomplete data, so it is not suitable for spatial autoregressive analysis. The second dataset is the complete distance matrix obtained from *amap.com* with the help of distance analysis tool provided by DataMap for Excel. However, it may not reflect the travel distance along the highways effectively. In computational process, the gravity model is based on the highway mileage table, while the spatial auto-regression is based on the distance matrix obtained from electronic maps (Table 1).

The algorithm is the multiple linear regression based on the least square method. Gravity model is a kind of nonlinear model, which cannot be implemented in regression analysis directly. Fortunately, it is easy to transform this model into a linear relation by taking logarithms of both sides. The nonlinear fitting method can also be used to estimate the parameters of this model directly. However, the disadvantage of curve fitting may lead to a result that the overall information of the dataset cannot be utilized effectively. The reasoning lies in that the bigger data points such as the city size of Wuhan has an excessive influence on the parameters. In particular, it is worth mentioning the size distribution of Hubei's cities follow what is called primate law rather than the rank-size law. Where urban population size is concerned, Wuhan stands head and shoulders above other cities in Hubei Province. In this case, the impact of Wuhan's population size on the model's parameter estimation based on curve fitting is overwhelming. After taking logarithm, the differences of city sizes lessen. It is an advisable selection to make use of multivariate linear regression analysis to estimate the parameter values of the local gravity model and the mixed gravity model with autoregressive term (S1 Dataset).

## 3 Results

### 3.1 Results and analysis of gravity modeling

The analytical process of geographical mathematical modeling involves three basic aspects. The first is mathematical structure, the second is model parameters, and the third is statistics such as correlation coefficient and standard errors. As for the mathematical structure, different model structures are determined by different distance decay functions. Whether spatial auto-regression is considered also affects model structure. The model parameters include constant term, size exponent and distance exponent or distance coefficients. If spatial auto-regression is considered, autoregressive coefficients should be included as well. The model statistics include

global statistics and local statistics. The global statistics include goodness of fit, *F* statistic and standard error of regression. The main local statistic is *t* statistic of each regression coefficient. Both the *F* statistic and the *t* statistic can be replaced by corresponding probability values (the *P* values or significance levels shown by statistical software). The probability values are very intuitive. If a *P* value is less than 0.05, it will pass the statistic test of 95% confidence level; if it is less than 0.01, it will pass the test of 99% confidence level. In this paper, the coefficient of variation is used in place of the standard error of regression. The calculation formula is: "coefficient of variation = standard error of regression/ the average value of logarithms of total number of confirmed patients". The *t* values of local statistics are replaced by corresponding probability values, namely, *P* values (S1 Table). For pure local gravity analysis, the parameters can be estimated by Eq (7). Highway mileage (the first type of road distance in this study) is adopted for it reflects the distance along the national highway and provincial highway, which is generally consistent with bus route. The distance within Wuhan cannot be defined as zero due to the logarithmic linear relation. The model can either ignore Wuhan itself, or estimate an internal diffusion distance. After repeated tests, it is found that 0.25 km is suitable for the internal distance of Wuhan city for local gravity modeling.

Let us examine the structure of the gravity model suitable for epidemic spread in Hubei at first. Model structure reflect the property at the macro level of spatial diffusion. The distance decay function of the gravity model given above follows inverse power law. In fact, the negative exponential function can be taken as the distance decay function of the gravity model as well. The mathematical properties of two decay functions are different. The exponential function bears a parameter indicating characteristic scale, while the power function has no parameter representing characteristic scale. The former implies simplicity, while the latter implies complexity [21]. As for the negative exponential decay, the spatial autocorrelation function of is a tailing-out curve, while the partial autocorrelation function is a cutoff-tailed curve. It implies that negative exponential decay essentially reflects a local action which lacks direct long range effect, and therefore it is inconsistent with the first law of geography. Both the autocorrelation function and partial autocorrelation function of inverse power law decay are tailing off, so it indicates the long range effect, which is consistent with the first law of geography. In the past, the spatial interaction model based on negative exponential function can be derived from the maximum entropy principle without involving dimensional problems [22, 23]. So it is favored by geographers. However, the negative exponential function have caused the contradiction between its locality and the first law of geography [24]. On the other hand, the gravity model based on inverse power law decay can be derived from the maximum entropy principle as well by changing the distance cost function [5, 25]. The dimensional problem can be solved by fractal geometry [5]. Therefore, both forms of gravity models are acceptable at now, and the selection between them should depend on the effect of empirical analysis. If we do not take logarithm of distance, i.e. using *r* instead of ln*r*, Eq (7) is equivalent to the gravity model whose distance decay function follows negative exponent rule. However, the statistic experiments show that the levels of confidence of the model parameters is relatively low and the fitting effect is poor in general when the negative exponential function is adopted. In contrast, the goodness of fit and the level of confidence of parameters of gravity model based on inverse power law are significantly better than model based on negative exponential function. After February 15, the model has tended to be stable. Taking February 16, 2020 as an example, the local gravity model of epidemic diffusion in Hubei Province can be expressed as

$$\hat{N}_j = 26.3327 P_j^{1.0135} r_j^{-0.4324}. \tag{9}$$

**Table 2. Comparison of goodness of fit of gravity models based on different distance decay functions.**

| Distance function | Autocorrelation | Partial autocorrelation | Space effect | Epidemic diffusion in Hubei Province |
|---|---|---|---|---|
| Negative exponential function | Tailed | Truncated | Quasi Locality: simple, limited scope, inconsistent with the first law of geography | Poor fitting effect |
| Negative power function | Tailed | Tailed | Long range effect: complex, unlimited scope, consistent with the first law of geography | Good fitting effect |

The goodness of fit $R^2 = 0.9507$, $F$ statistic is 134.9169, and the corresponding probability value sig. $< 0.0001$. The coefficient of variation $\delta = 0.0580 < 0.1$, and the degrees of confidence of all parameters were greater than 99.96%. Referring to the previous calculation results, the mathematical model expression of each date can be presented (S1 Table). The results imply that the diffusion of COVID-19 in Hubei Province obeys the inverse power law indicative of action at a distance. The diffusion process does not well accord with negative exponential law so it is not a local spatial diffusion process (Table 2).

Secondly, the model parameters should be investigated to see the properties at the micro level of spatial diffusion. The gravity coefficient of the local gravity model contains the information of the population size of the central city, Wuhan. The value of the gravity coefficient has gradually increased, indicating that the epidemic situation of Wuhan was becoming more and more serious. The size exponent firstly decreased, and then rebounded after January 30. After February 2, the size exponent value gradually became larger than one. After February 15, it showed a downward trend again and the value became less than one after February 20. The variation implies that the impact of city size was not very prominent before February 2, and later the effect of urban population size needed to be highlighted. After February 15, the human-to-human transmission occurred less frequently because of isolation measures taken by local governments, and the scale effect declined after February 20. The distance exponent also showed a downward trend before February 1, and then gradually increased. The exponent rose sharply on February 12 (Fig 2). This variation implies slow distance decay and rapid spatial diffusion before February 1. After February 1, the space isolation measures taken by local governments gradually took effect, slowly increasing the impedance of distance. Especially after February 12, the space diffusion became more difficult than before (Table 3). Overall, the gravity coefficient reflects the epidemic situation and its impact of the central city, Wuhan. The distance exponent reflects the distance from the central city and the impact of direct transmission from Wuhan, while the size exponent reflects the size of other cities and the impact of local secondary transmission of the epidemic.

Finally, the model statistics are analyzed for testing modeling effect. Model statistics are statistical measurements which can be used to evaluate the mathematical structure of a model or the level of confidence of parameter values. An absolutely reliable model statistic does not exist in this world. A model which cannot pass the statistical test usually has some problems. However, a model which does have some problems may also pass the statistical test. Different statistic values should be taken into account together in statistical test. From the perspective of global statistics, both the correlation coefficient square $R^2$ and $F$ statistic decreased before January 31, and then showed a gradual upward trend. These two statistics rose rapidly after February 13, and then declined slightly after February 16. The coefficient of variation kept declining without significant fluctuations. The square of the correlation coefficient reflects the goodness of fitting of the gravity model to the observed data and the degree of explanation of the logarithms of the distances and city sizes for the logarithm of the total number of confirmed cases. The coefficient of variation indicates the linear prediction accuracy of the model. Empirically, the coefficient of variation should be less than 0.1. The accuracy of prediction has reached the

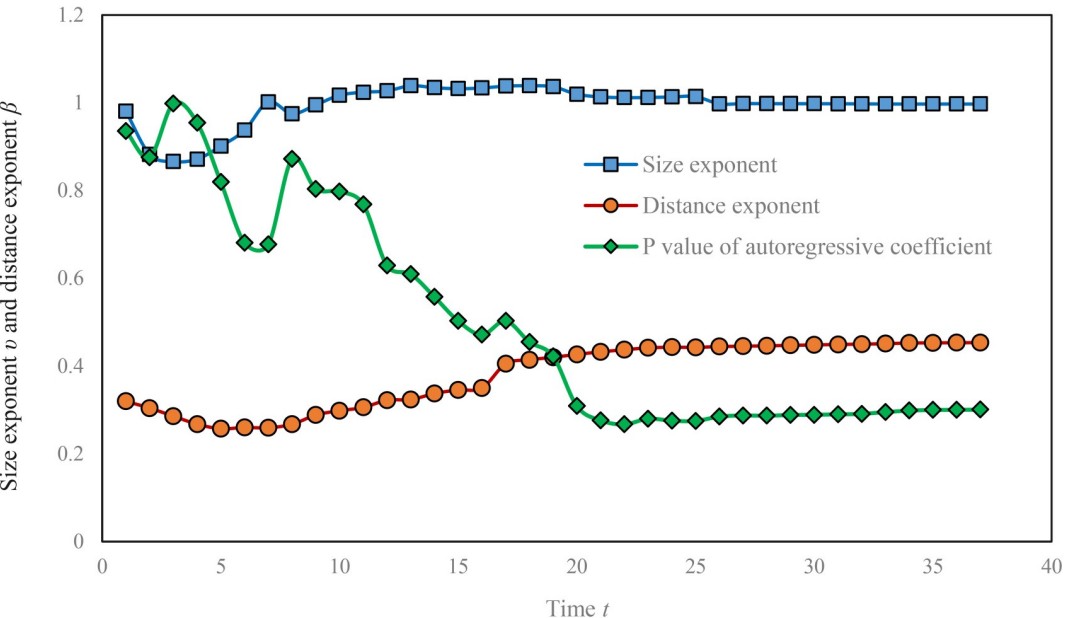

**Fig 2. The variation of gravity scaling exponent of COVID-19 diffusion in Hubei Province. (1)** The first point of the sequence of horizontal axis represents January 28. Previous data were not complete enough to be used. **(2)** The size exponent and distance exponent are estimated based on Eq (7), and the probability value of autoregressive coefficient is given based on Eq (8).

standard since January 31. $F$ statistic is used to test whether there is at least one of two measurements, logarithm of city size and logarithm of distance, can effectively explain the logarithm of total number of confirmed cases. The changing trend of $F$ statistic is generally consistent with the trend of above parameters. The main local statistic are the probability values corresponding to $t$ statistics, which reflect whether gravitational coefficient and scaling exponent, which also known as cross elasticity coefficient, are significantly different from 0. No significant difference between a certain elasticity coefficient and 0 implies that the variable has no significant influence on the total number of confirmed cases or the calculation error is obvious. The probability value of the logarithm of the gravitational coefficient showed an overall downward trend. The confidence level reached 90% on January 31, and then reached 95% on February 4, and finally exceeded 99.9% after a gradual increase. Most confidence levels of size exponent and distance exponent are above 99%. Although the level of confidence of distance exponent on January 30 is relatively low, it is not less than 95%. The probabilities of the

**Table 3. The meanings, variation characteristics and geographical phenomena of parameters of gravity model of COVID-19 spread in Hubei Province.**

| Parameter | Property | Meaning | Variation characteristics | Geographic information |
|---|---|---|---|---|
| Gravitational coefficient $\eta$ | Central scale factor | In proportion to the confirmed cases of COVID-19 in central cities | Showed an upward trend on the whole and dropped occasionally on February 3. | The number of people infected in the central city, Wuhan was rising rapidly. |
| Size exponent $\upsilon$ | Size exponent: secondary transmission effect | The relative share of the increased confirmed cases of COVID-19 which corresponds to the relative share of increased city size. | Decreased before February 1, then increased gradually, and decreased again after February 14. | After February 1, the size of cities has a prominent effect on the transmission of COVID-19, implying secondary transmission |
| Distance exponent $\beta$ | Distance scaling exponent: primary transmission effect | The relative share of the increased confirmed cases of COVID-19 which corresponds to the relative share of decreased distance from Wuhan. | Declined before February 1 and then increased gradually. | Rapid diffusion directly from Wuhan before February 1 and since then barriers to epidemic diffusion have increased, implying local isolation. |

two exponents rose slightly before February 1 and kept declining thereafter. There are two possible explanation of the variation of parameters and statistics: one is that the caliber of statistics has been adjusted, and the other is that the system evolution is not stable enough. The problem is that January 31, 2020 is a time node from all perspectives. One possibility is that the epidemic diffusion process reached a stable state in Hubei around January 31 and the diffusion pattern was formed at that time.

## 3.2 Spatial autoregressive analysis based on gravity model

In order to investigate whether there is cross transmission of COVID-19 between different regions of Hubei Province, spatial autoregressive analysis can be carried out. Eq (8) can reflect the assumption that epidemic have spread through the interaction between different regions. Because complete distance matrix cannot be built from the highway mileage data provided by atlas, the data of road distances between cities are extracted from *amap.com* by means of GIS technology to build complete distance matrix. The internal distance of Wuhan is taken as 0.25 km, which is consistent with the analysis of gravity model. Firstly, it is necessary to take reciprocal of all values in the distance matrix to generate the spatial proximity matrix. Specially, let diagonal elements to be 0. Then, the spatial proximity matrix is globally unitized (normalized) to be transformed into a spatial weight matrix. After that, the $W\ln N$ is obtained by multiplying the spatial weight matrix and the vector of logarithm of the total number of confirmed cases. Finally, taking $\ln r$, $\ln P$ and $W\ln N$ as independent variables and $\ln N$ as dependent variables, we can carry out multiple linear regression to obtain the parameters of the model and corresponding global and local statistics. The results show that the value of $t$ statistics for the autoregressive coefficient is too low, so the corresponding probability ($P$-value) is too high. $P$ value increased at first and then decreased. The highest value is 0.998, and the corresponding confidence level is only around 0.2%. The lowest value is 0.2672, and the corresponding confidence level is around 73.28%. After February 16, the $P$-value corresponding to autoregressive coefficient tends to be stable, varying around 0.285, and the corresponding confidence level is around 71.5%. In order to be intuitive, the probability $P$-value of autoregressive coefficient is added to Fig 3. The conclusion can be reached that the trend of horizontal transmission of COVID-19 in Hubei Province is not significant. However, the trend was gradually strengthened before February 16. Taking February 16 as an example, the linear expression of the local gravity model with autoregressive term is as follows

$$\ln \hat{N}_j = 2.8802 + 1.0135 P_j - 0.4075 r_j + 0.6485 W\ln N_j. \tag{10}$$

The goodness of fit $R^2 = 0.9517$, $F$ statistic is 85.3619 and its corresponding probability *sig.* <0.0000. The coefficient of variation $\delta = 0.0596 < 0.1$, and the level of confidence of intercept is more than 99.7%. The confidence levels of size exponent and distance exponent are more than 99.995%, but the confidence level of autoregressive coefficient is less than 73%. Due to different degrees of freedom, the goodness of fit of Eqs (9) and (10) is not comparable with one another. However, the adjusted goodness of fit $R_{adj}^2$ is comparable. The former is $R_{adj}^2 = 0.9436$, and the latter is $R_{adj}^2 = 0.9404$. $F$ statistic and coefficient of variation are comparable. The comparison shows that the interpretation effect and prediction effect of the gravity model are not improved after introducing the autoregressive term (S2 Table).

The insignificance of spatial autoregressive coefficient indicated that the spatial autocorrelation of epidemic transmission in Hubei Province was not significant. The experiment shows that the spatial autocorrelation is indeed insignificant in Hubei Province (Limited to the space of the paper, the relevant issues will be discussed separately). After the outbreak of the

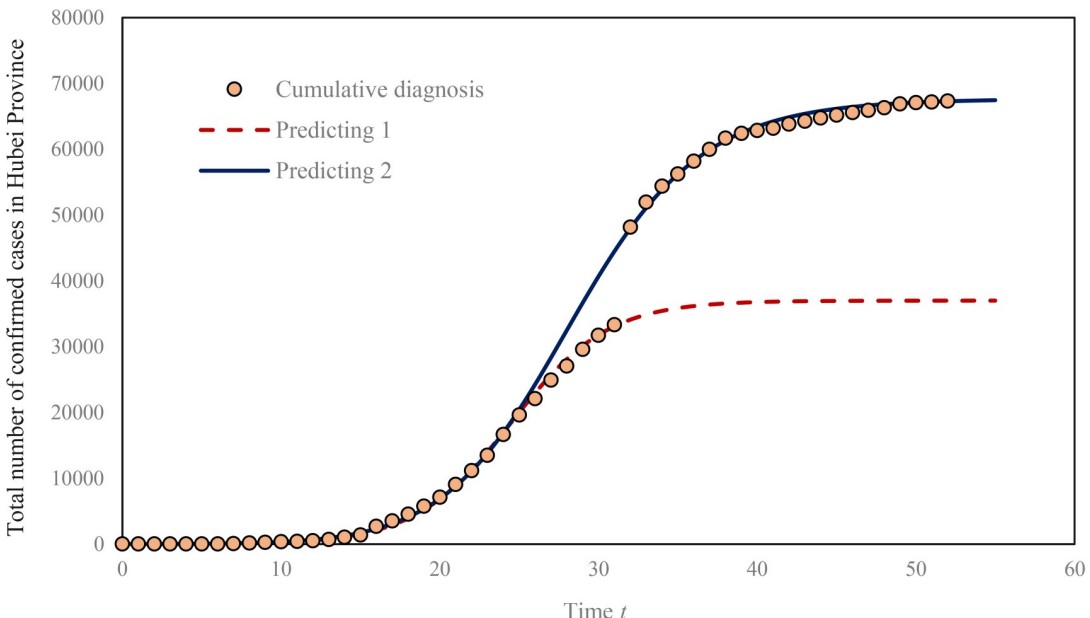

**Fig 3. The sigmoid growth curve and the logistic process of total number of confirmed cases of COVID-19 in Hubei Province.** Due to the adjustment of the diagnosis standard of COVID-19, the statistical caliber has varied several times. The most significant change took place on February 12, 2020. Statistically speaking, the growth rate of the total number of confirmed cases on February 12 is an abnormal value. From February 19 to 26, the statistical caliber seemed to be changeable again for the value is abnormal to some extent. Here are two trend lines based on two sets of data with different statistical calibers. The first logistic curve is more consistent with the growth rate, and the second fractional logistic curve can reflect the capacity of confirmed cases. The bifurcation point of these two curves is around February 4 and 5. The growth rate of COVID-19 reached its peak on these two days.

epidemic, local governments may take necessary lockdown and isolation measures [26, 27]. There are different opinions on how to evaluate these measures. The results of spatial autoregressive analysis show that, on the one hand, the isolation measures taken by central and local governments have played an effective role; on the other hand, the spatial interaction still exists to some extent. Based on confidence level of more than 80%, the horizontal relationship of transmission of COVID-19 can be ignored. It can be judged that the spatial autocorrelation of COVID-19 spread in Hubei should have been stronger without the strong spatial isolation measures taken by the government. It was the space isolation measures that curbed the disordered spread of COVID-19 in Hubei Province and China.

## 4 Discussion

The form of a mathematical model reflects the macro structure of a system, while the parameters of the model reflects the characteristics of the micro element correlation or interactions of the system. Accordingly, the statistics (e.g., $R^2$, $F$, $t$) of fitting the model to observed data tell us whether and to what extent the model and its parameters are convincing from different perspectives. In light of the global statistics (e.g., $R^2$, $F$) and local statistics (e.g., $t$), it is generally credible to use the gravity model to describe and explain the transmission of COVID-19 in Hubei within the study period and area. At the beginning, the credibility of gravity coefficient is relatively low due to the unstable state of epidemic diffusion and the obvious random disturbance, thus the model could not effectively reflect the epidemic situation in Wuhan. After February 1, 2020, the model became more and more stable and the goodness of fit became better and better. The comprehensive results of model selection, parameter analysis and statistical

test can be summarized as follows. *Firstly, the epidemic transmission in Hubei has long-range effect at the macro level.* Therefore, it can be judged that the COVID-19 is able to spread to an infinite distant place theoretically. If the epidemic diffusion is localized, the goodness of fit of the gravity model based on negative exponential attenuation would be better that that based on inverse power law. In fact, the gravity model based on inverse power law is more consistent with the observed data. After February 1, the distribution pattern of COVID-19 caused by gravity effect tended to be stable. *Secondly, the effects of city size and spatial distance on epidemic situation are different in different periods at the micro level.* Before February 1, the distance decay exponent was low, indicating that the distance from Wuhan to other cities played a major role in COVID-19 diffusion. In contrast, the size exponent is less than 1, indicating that the role of population size of different regions is not prominent. After February 1, the distance exponent gradually increased, rising significantly on February 13. The distance exponent reflects the direct impact of Wuhan on other regions. The exponent changed implies that space isolation had taken significant effect since February 1, and the effect of space isolation was more obvious on February 13. At the same time, the size exponent increased, indicating that the size of each city came into play. The population size is more related to the secondary transmission, so the increasing significance of population size reflects that the direct transmission from Wuhan was gradually turned into the secondary transmission within every region. *Thirdly, the horizontal transmission of the epidemic between other cities is not significant.* The level of confidence of spatial autoregressive coefficient is not high. Although the confidence level of the autoregressive coefficient gradually increased over time, it failed to reach 85% throughout the research period. This reflects the significant effect of space isolation measures taken by government of China from another perspective. Otherwise, there would be significant spatial autocorrelation between different regions of Hubei, and thus the autoregressive coefficients of the mixed gravity model should be significant.

The gravity models are based on distance decay law which can be expressed by a certain type of impedance function. In geography, the impedance functions are also termed distance decay functions, which are not limited to negative exponential function and inverse power function. General typology of distance decay functions include power function, exponential function, square root exponential function, normal function, lognormal function, gamma function, and so on [3, 8, 9, 28, 29]. In fact, linear function and logarithmic function can also serve as distance decay functions. These distance decay functions fall into two categories: one is those that can be directly derived from the principle of entropy maximization, including linear decay function, exponential function, normal distance, and power function; the other is those that cannot be directly derived by means of the entropy-maximizing method, including logarithmic function, lognormal function, square root exponential function, and gamma function (Table 4). Among various distance decay functions, the inverse power function is special. On the one hand, it can be derived from double entropy maximization processes, and on the other, it has no characteristic scale (Chen, 2008). In this work, the main ideas of choosing distance decay functions are as follows. *First, the principle of spatial entropy maximization.* Geographical gravity and spatial interaction modeling are associated with the process of maximizing entropy [5, 22, 23, 25, 30]. In this sense, only linear function, exponential function, power function and normal function are acceptable for gravity modeling. *Second, the notion of spatial complexity.* Geographical systems are complex spatial systems [31–34]. Generally speaking, complex systems bear no characteristic scales and do not satisfy linear spatial process [35]. Therefore, linear functions can be excluded. *Third, spatial diffusion characteristics.* Distance decay results in special spatial distribution states. Spatial distribution and probability distribution are essentially two different aspects of the same problem. There are three probability density distribution functions which are associated with the principle of entropy

**Table 4. The common distance decay functions in geographical analyses and its mathematical and physical properties.**

| Type | Name | Function | Entropy maximization | Characteristic scale |
|---|---|---|---|---|
| Linear | Linear Model: | $T(r) = T_0 - kr$ | Yes | **Yes** |
| Single logarithm | Normal | $T(r) = T_0\exp(-kr^2)$ | Yes | Yes |
| | Exponential | $T(r) = T_0\exp(-kr)$ | No | Yes |
| | Square root exponential | $T(r) = T_0\exp(-kr^{1/2})$ | No | Yes |
| | Logarithmic | $T(r) = T_0 - k\ln r$ | No | Yes |
| Double logarithms | Power law | $T(r) = T_1 r^{-\alpha}$ | Yes | No (scaling) |
| | Lognormal | $T(r) = T_1\exp(-k(\ln r)^2)$ | No | Yes |
| Mixed | **Gamma** | $T(r) = T_1 r^{-\alpha}\exp(-kr)$ | **No** | Yes |

**Notes**: (1) Notation: $T$ = Interaction between two locations; $r$ = Distance between two locations; $T_0$, $T_1$, $k$, $\alpha$ = Constants; $e$ = Exponential constant (2.7183). (2) References: [3, 8, 10, 28]. (3) Properties: the relationships between the distance decay functions and entropy maximization and characteristic scale are summarized in this study.

maximization. That is, uniform distribution (linear function), negative exponential distribution (exponential function), and normal distribution (Gaussian function) [35]. If the distance variable $r$ has a clear lower limit $a$ and an upper limit $b$ (i.e., $-a<r<b$, and $a\geq0$, $b>0$ represent constants), the probability density based on entropy maximization satisfies the uniform distribution and can be described by a linear function; If the distance variable $r$ has a clear lower limit $r = 0$, but no upper limit (i.e., $0\leq r<\infty$), the entropy-maximization-based probability density meets the exponential distribution and can be modeled by an exponential function; If the distance variable $r$ has neither a lower limit nor an upper limit (i.e., $-\infty<r<\infty$), the probability density based on entropy maximization takes on the normal distribution [36, 37]. Spatial diffusion starts from a certain center and spreads to the periphery, which belongs to the second case mentioned above. A pair of exponential decay processes compose a power law decay process [35]. So, the normal function can be excluded in this work, despite the fact that it is useful in geographical spatial analysis [29]. Finally, only negative exponential function and inverse power law function can be utilized in the local gravity modeling.

The characteristics of certain temporal process could be reflected from the pattern of spatial diffusion, for the spatial pattern and the temporal process depend on each other. The growth of confirmed cases of an epidemic usually appears as an S-shaped curve [4, 38]. The simplest S-shaped curve is logistic curve. The growth of confirmed cases of COVID-19 may increase exponentially in the short term [39], but in the long run it is logistic growth [38]. A process of logistic growth can be divided into three or four stages according to the velocity and acceleration [40–42]. The *initial stage*, the *celerity stage* and the *terminal stage* represent three stages of a logistic curve. These three stages correspond to the three stages of spatial diffusion process, which are the *primary stage*, the *diffusion stage*, the *shrinking stage* or *saturation stage* [4, 7]. S-shaped curves, such as logistic curves, correspond to the squashing function. The first derivative of the squashing function gives velocity curve with one peak, while the second derivative gives an acceleration curve with one peak and one valley [43]. In this way, an S-shaped curve can be divided into four stages: the *initial stage*, the *acceleration stage*, the *deceleration stage* and the *terminal stage* [40]. Correspondingly, the spatial diffusion process can be divided into four stages as well: the *primary stage*, the *spatial diffusion stage*, the *hierarchical diffusion stage* and the *shrinking stage* (Table 5). In fact, the spatial diffusion stage is a process dominated by direct transmission from the diffusion center like Wuhan, while the hierarchical diffusion stage is a process dominated by secondary transmission from the central parts of other regions. The importance of two kinds of diffusion patterns in two stages are different, which cannot be strictly distinguished from one another. The epidemic situation in Hubei Province is generally

**Table 5. The stage division and comparison of time and space diffusion processes.**

| Stage | Three stage division | | Four stage division | |
|---|---|---|---|---|
| | Spatial diffusion | Temporal growth | Spatial diffusion | Temporal growth |
| Stage 1 | Primary stage | Initial stage | Primary stage | Initial stage |
| Stage 2A | Diffusion stage | Celerity stage | Spatial diffusion stage | Acceleration stage |
| Stage 2B | Diffusion stage | Celerity stage | Hierarchical diffusion stage | Deceleration stage |
| Stage 3 | Shrinking stage | Terminal stage | Shrinking stage | Terminal stage |

**Note:** (1) Northam called the second stage of S-shaped curve acceleration stage [42], which is not accurate because the curve in this stage accelerate first and then decelerate. (2) The terminal stage of spatial diffusion is also called saturation stage.

characterized by S-shaped curve (Fig 3). The growth peak of the total number of confirmed cases of COVID-19 was approximately February 4 or February 5, and, in theory, the most severe period was from January 27 to February 15. Therefore, the epidemic situation in Hubei Province can be roughly divided into four stages.

In geographical analysis, it's natural to use gravity model to study the pattern and process of diffusion of epidemic. However, the objective of this paper lies chiefly at the gravity scaling analysis based on allometric relation, gravity model and spatial autoregressive analysis. In a sense, this paper is devoted to developing a local gravity model to describe the spatial diffusing processes based on core-periphery patterns. The main task of sciences is to make models [44], and models reflect the quintessential human activity [45]. The basic role of models is to explain and predict [13, 46, 47]. However, models are not universally applicable. All mathematical models have their own scope of application. The principal applicable conditions of the local gravity models can be summarized as follows. (1) Generally, the models can be applied to the single center diffusion process based on core-periphery interaction. (2) The relation between the number of confirmed cases and passenger flow follow the allometric scaling law. (3) The model based on negative exponential decay is suitable for the spatial process with locality (simpler spatial systems with characteristic scale), while the model based on inverse power law decay is suitable for the spatial process of long-range action (complex spatial systems without characteristic scale). Despite these limitations, maybe the model can be applied to core-periphery diffusion process in other fields such as information technology diffusion. The diffusion of the movable type printing press is an interesting phenomenon [48, 49]. Likely, this technology diffusion process and patterns can be characterized by the local gravity model.

The main academic contribution of this paper is to develop the spatial diffusion model based on local gravity. Based on the allometric scaling relation, the local gravity model about the passenger flow is transformed into the local gravity model about the case transmission quantity. A series of COVID-19 papers published have been published, showing many interesting research results. However, in terms of research objectives, results and nature, this paper is quite different from the published papers. The application of mathematical tools in scientific research has two main functions: one is to sort out observational data, and the other is make theoretical models. In the previous works about COVID-19, mathematics was mainly employed to process data, while this paper is to develop theoretical model of spatial diffusion. The shortcomings of this study are as follows. The first is the limitations of data. The urban population size is obtained from the statistical data collected two years ago, and the total number of confirmed cases may not be completely accurate. The second is the limitations of the research area. This paper takes the administrative boundary of Hubei Province as the boundary of study area, but the spatial transmission of epidemic cannot be prevented by the administrative boundary. The third is the limitation of time. Since log linear regression analysis is used

in this paper, the values must be larger than 0. So the dates before January 28 were not included in the time series analysis of parameters due to the incomplete datasets and the difficulties in comparing. Despite all these deficiencies, the general trend of the COVID-19 is very clear. The day-by-day comparison showed that the modeling results based on epidemic data not only reflected certain spatial regularities, but also revealed significant geospatial information. Further study could consider breaking through the administrative boundary and analyzing epidemic transmission from Wuhan in a larger research area.

## 5 Conclusions

As a basic mathematical method of geospatial analysis, gravity model is useful in researching the temporal and spatial characteristics of epidemic transmission. The local gravity model for spatial diffusion was derived from the global gravity model and the allometric scaling relation between passenger flow and case transmission quantity. Which differs from the traditional local gravity model which is suitable for passenger flow. Then the newly derived local gravity model was employed to analyze the spatial diffusion from Wuhan to other cities in Hubei, and the spatial auto-regression based on the gravity model was used to investigate whether there is interaction between all prefecture-level cites in Hubei Province. The analysis results reflect the quality of the models. This study demonstrated that the spatial diffusion process of COVID-19 in Hubei Province of China was dominated by gravitational rule. Through the comparison of model structure, the analysis of the model parameter change and the variation of model statistics, chief conclusions about spatial diffusion of COVID-19 in Hubei Province can be drawn as follows.

### Firstly, the COVID-19 diffusion from Wuhan to other regions in Hubei Province bears long-range effect

This can be judged by the distance decay functions. According to the matching analysis of the local gravity model with observation data, the goodness of fit of the gravity model based on power-law distance decay is better than that of the gravity model based on negative exponential distance decay. This indicates that the diffusion mechanism of COVID-19 is complex and the spread is not limited within a clear boundary. That is, in theory, COVID-19 can spread from Wuhan to an infinitely distant place. The reality is that COVID-19 once spread almost all over China from the central city, Wuhan. This conclusion may be superfluous, but it is of some significance. It proves that the epidemic diffusion is not localized, and the gravity model based on negative exponential function cannot well describe the spatio-temporal evolution of epidemic effectively.

### Secondly, the speed and the scale of spatial diffusion process of COVID-19 from Wuhan mainly depend on city sizes and the distances from Wuhan

This can be judged by the mathematical structure of the local gravity model. The total number of confirmed cases of COVID-19 is directly proportional to a certain power of population size, and inversely proportional to a certain power of distance. Before February 1, the impact of distance from Wuhan was prominent, which indicated that the epidemic situation in Hubei Province was mainly caused by direct transmission from Wuhan, namely, spatial diffusion process. After February 1, 2020, the effect of city size became increasingly prominent while the distance decay exponent gradually increased. This indicated that the speed of direct transmission from Wuhan to other places had been controlled to a certain extent due to the isolation measures taken by Chinese government. The effect of the cardinal number of infected people in each

region is manifested, and the secondary diffusion, namely, hierarchical diffusion mode, have emerged since February 1, 2020.

### Thirdly, the vertical transmission directly from Wuhan to other regions and the secondary transmission within every region are the main diffusion modes of COVID-19 in Hubei Province, while the horizontal cross-transmission between different regions is not significant

This can be judged by the results of spatial auto-regression analysis. The spatial autoregressive term is introduced into the gravity model, but the autoregressive coefficient is not significant. In general, the significance of autoregressive coefficient has gradually increased, but the confidence level failed to reached 80%. The result shows that due to the timely isolation measures, the interaction of COVID-19 spread between different regions in Hubei has not played an important role. This also demonstrates that the isolation measures taken by the government of China are effective although some people regarded "lockdown" as a clumsy method. Geospatial isolation is a necessary means to prevent and control the diffusion of COVID-19 when vaccines and efficacious drugs against the virus have not been developed yet.

## Supporting information

**S1 Table. Parameters of gravity model and corresponding statistics of diffusion process of COVID-19 in Hubei Province.** The main results of multiple linear regression based on Eq (7) for local gravity modeling, including estimated parameter values and statistics, are listed in the table for reference and comparison.
(DOCX)

**S2 Table. Gravity model parameters with spatial autoregressive terms and their corresponding probability values.** The main results of multiple linear regression based on Eq (8) for the local gravity modeling associated with spatial autoregressive process are listed in the table for reference and comparison.
(DOCX)

**S1 Dataset. Datasets for spatial analysis of COVID-19 diffusion in Hubei Province, China.** This contains the original data and calculation process for the gravitational scaling analysis on spatial diffusion of COVID-19 in Hubei Province, China. The data are follows: (1) Population sizes of all the cities in Hubei Province. (2) Spatial distance vectors and matrix. (3) Cumulative number of novel coronavirus pneumonia confirmed at different dates.
(XLSX)

## Acknowledgments

I would like to thank the two anonymous reviewers and Dr. Jing Sun whose interesting and constructive comments were very helpful in improving the quality of this paper.

## Author Contributions

**Conceptualization:** Yanguang Chen.

**Data curation:** Yanguang Chen, Yajing Li, Xiaoming Man.

**Formal analysis:** Yanguang Chen.

**Funding acquisition:** Yanguang Chen.

**Investigation:** Yanguang Chen, Yuqing Long.

**Methodology:** Yanguang Chen.

**Project administration:** Yanguang Chen.

**Resources:** Yanguang Chen.

**Software:** Yajing Li, Xiaoming Man.

**Supervision:** Yanguang Chen.

**Validation:** Yanguang Chen.

**Visualization:** Yajing Li.

**Writing – original draft:** Yanguang Chen.

**Writing – review & editing:** Yanguang Chen, Yajing Li, Shuo Feng.

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
