## [Decision Letter · Decision Letter 0]

20 Apr 2021

PONE-D-21-07713

Gravitational Scaling Analysis on Spatial Diffusion of COVID-19 in Hubei Province, China

PLOS ONE

Dear Dr. Chen,

Thank you for submitting your manuscript to PLOS ONE. After careful consideration, we feel that it has merit but does not fully meet PLOS ONE’s publication criteria as it currently stands. Therefore, we invite you to submit a revised version of the manuscript that addresses the points raised during the review process.

We look forward to receiving your revised manuscript.

Kind regards,

Bing Xue, Ph.D.

Academic Editor

PLOS ONE

Journal Requirements:

"This research was sponsored by the National Natural Science Foundation of China (Grant No.

41671167). The support is gratefully acknowledged."

" The funders had no role in study design, data collection and analysis, decision to publish, or preparation of the manuscript."

3. We note that Figure 1 in your submission contain map images which may be copyrighted. All PLOS content is published under the Creative Commons Attribution License (CC BY 4.0), which means that the manuscript, images, and Supporting Information files will be freely available online, and any third party is permitted to access, download, copy, distribute, and use these materials in any way, even commercially, with proper attribution. For these reasons, we cannot publish previously copyrighted maps or satellite images created using proprietary data, such as Google software (Google Maps, Street View, and Earth). For more information, see our copyright guidelines: http://journals.plos.org/plosone/s/licenses-and-copyright.

3.1.    You may seek permission from the original copyright holder of Figure 1 to publish the content specifically under the CC BY 4.0 license. 

3.2.    If you are unable to obtain permission from the original copyright holder to publish these figures under the CC BY 4.0 license or if the copyright holder’s requirements are incompatible with the CC BY 4.0 license, please either i) remove the figure or ii) supply a replacement figure that complies with the CC BY 4.0 license. Please check copyright information on all replacement figures and update the figure caption with source information. If applicable, please specify in the figure caption text when a figure is similar but not identical to the original image and is therefore for illustrative purposes only.

Reviewers' comments:

Reviewer's Responses to Questions

**Comments to the Author**

1. Is the manuscript technically sound, and do the data support the conclusions?

Reviewer #1: Yes

Reviewer #2: Yes

Reviewer #3: Yes

2. Has the statistical analysis been performed appropriately and rigorously? 

Reviewer #1: Yes

Reviewer #2: Yes

Reviewer #3: Yes

3. Have the authors made all data underlying the findings in their manuscript fully available?

Reviewer #1: Yes

Reviewer #2: Yes

Reviewer #3: Yes

4. Is the manuscript presented in an intelligible fashion and written in standard English?

Reviewer #1: Yes

Reviewer #2: Yes

Reviewer #3: Yes

5. Review Comments to the Author

Reviewer #1: It is a good paper with a clear presentation of the model, its implementation in modeling the pandemic, and a thorough interpretation of the results.

My comments are minor and discretionary:

1. The distance decay functions employed by the study include power and exponential functions. I understand that the two are the most popular ones used in the literature. The literature also suggests additional functions such as square-root exponential and log-normal (Taylor, 1983, Distance decay in spatial interactions. In Concepts and Techniques in Modern Geography) and Gaussian (Shi et al. 2012, Annals of AAG 102, 1125-1134). Future work may expand the selection set.

2. On interpretation/discussion of the result, I am not surprised by the strong effect of Wuhan on peripheral cities and no much between peripheral cities themselves, which is consistent with the spatial interaction pattern in passenger flows (even before the pandemic) in Hubei. In other words, if actual passenger flows could be obtained in the region (which is not easy in China), one could predict/explain the pandemic by that observed data without modeling the interaction.

3. I like the information presented in Table 2 and Figure 2, which are considered the main results of the paper. However, Table 2 seems to be redundant as Figure 2 tells the same info with better visual effect.

4. The lockdown in late Jan in the region completely brought population movement to a stop. I'd suspect that the temporal lag in the diagnosis cases in various cities reflects the incubation period.

Reviewer #2: This research proposes an integrated mathematical model combining gravity model with spatial autoregression model to investigate the spatial diffusion of COVID-19 in Hubei province, China. Generally, this manuscript provides us sufficient information about model selection and establishment, data source, and statistical test. The effectiveness of the proposed method was tested empirically and statistically. The analytical results are of certain enlightening value for understanding the spatial diffusion process of COVID-19, especially in terms of the effects of population size and spatial distance. However, to be accepted for publication, the presentation still need improvement. Detailed comments are as follows:

The mathematical model proposed in this study is suitable for certain conditions. Specifically, the gravity model describes the one-directional transmission of COVID-19 from Wuhan to other cities, and the spatial autoregression model considers horizontal diffusion between cities (except Wuhan) within Hubei province. More details about the epidemic prevention and control measures (i.e., spatial isolation of cities and the corresponding time spots of the implementation) should be provided to illustrate that the practical situation in Hubei is consistent with applicable conditions of the model.

Reviewer #3: Comments to the Author

The authors used gravitational scaling analysis to study the spatial transmission of COVID-19 in Hubei, China. I provided my feedbacks based on my first reading and my experiences.

Abstract:

The manuscript needs to explore the results more deeply. I don’t think the main result in the Abstract “The local gravity model based on power law decay can effectively describe the diffusion patterns and process of COVID-19 in Hubei Province, and the goodness of fit of the gravity model based on negative exponential decay to the observational data is not satisfactory” is informative, or useful. The authors only described the model function, not the COVID-19 transmission. Most readers are not interested in the goodness of fit or confidence level, which should be within the acceptable level, as default.

Conclusion part in the abstract is more important, which should replace the current results part. Yet, the authors need to clarify some information here: what does long-range effect mean, please explain? “the size of a city and the distance …affect the total number of confirmed cases”, positively affect or negatively affect, or just affect? What are direct and horizontal transmission?

The authors need definitely rewrite the abstract. Please refer to some journals’ guidance, like landscape ecology, whose abstract is structured.

Main text:

Since 2020, there have been a series of COVID-19 papers published, many of which focus on spatial transmission, like epidemiology study. The authors need to make a brief review/comparison of these publications, since the authors’ stance is from the urban geography perspective, your strengthen?

Section 3.1 should not place in the Results part.

Table 2. I cannot directly find useful information from this long table. Suggestion, plot some columns?

The authors’ work focuses on Wuhan, the first city attacked by the COVID-19 as reported. It would be useful to make a comparison study, with other provinces (sub-nation unit) in China or other countries using the model.

OVERALL

The paper provides an important perspective in studying COVID-19 transmission, which is useful and has potential implications in policy making. I hope that my feedback comments are useful to help improve it in this regard.

6. PLOS authors have the option to publish the peer review history of their article (what does this mean?). If published, this will include your full peer review and any attached files.

Reviewer #1: No

Reviewer #2: No

Reviewer #3: **Yes: **Jing Sun

---

## [Decision Letter · Decision Letter 1]

25 May 2021

Gravitational Scaling Analysis on Spatial Diffusion of COVID-19 in Hubei Province, China

PONE-D-21-07713R1

Dear Dr. Chen,

We’re pleased to inform you that your manuscript has been judged scientifically suitable for publication and will be formally accepted for publication once it meets all outstanding technical requirements.

Kind regards,

Bing Xue, Ph.D.

Academic Editor

PLOS ONE

Additional Editor Comments (optional):

Reviewers' comments:

Reviewer's Responses to Questions

**Comments to the Author**

1. If the authors have adequately addressed your comments raised in a previous round of review and you feel that this manuscript is now acceptable for publication, you may indicate that here to bypass the “Comments to the Author” section, enter your conflict of interest statement in the “Confidential to Editor” section, and submit your "Accept" recommendation.

Reviewer #1: All comments have been addressed

Reviewer #3: All comments have been addressed

2. Is the manuscript technically sound, and do the data support the conclusions?

Reviewer #1: Yes

Reviewer #3: Yes

3. Has the statistical analysis been performed appropriately and rigorously? 

Reviewer #1: Yes

Reviewer #3: N/A

4. Have the authors made all data underlying the findings in their manuscript fully available?

Reviewer #1: Yes

Reviewer #3: (No Response)

5. Is the manuscript presented in an intelligible fashion and written in standard English?

Reviewer #1: Yes

Reviewer #3: Yes

6. Review Comments to the Author

Reviewer #1: I am happy with the revision.

I recommend for its acceptance.

I am happy with the revision.

I recommend for its acceptance.

Reviewer #3: (No Response)

7. PLOS authors have the option to publish the peer review history of their article (what does this mean?). If published, this will include your full peer review and any attached files.

Reviewer #1: **Yes: **Fahui Wang

Reviewer #3: No

---

## [Editor Report · Acceptance letter]

4 Jun 2021

PONE-D-21-07713R1 

Gravitational Scaling Analysis on Spatial Diffusion of COVID-19 in Hubei Province, China 

Dear Dr. Chen:

I'm pleased to inform you that your manuscript has been deemed suitable for publication in PLOS ONE. Congratulations! Your manuscript is now with our production department. 

Kind regards, 

on behalf of

Professor Bing Xue 

Academic Editor

PLOS ONE